# Protective Effect of Leisure-Time Physical Activity and Resistance Training on Nonalcoholic Fatty Liver Disease: A Nationwide Cross-Sectional Study

**DOI:** 10.3390/ijerph19042350

**Published:** 2022-02-18

**Authors:** Jae Ho Park, Nam-Kyoo Lim, Hyun-Young Park

**Affiliations:** 1Division of Population Health Research, Department of Precision Medicine, Korea National Institute of Health, Korea Disease Control and Prevention Agency, 200 Osongsaengmyeong2-ro, Osong-eup, Heungdeok-gu, Cheongju-si 28160, Chungcheongbuk-do, Korea; pjh666@korea.kr (J.H.P.); nklim@korea.kr (N.-K.L.); 2Department of Precision Medicine, Korea National Institute of Health, Korea Disease Control and Prevention Agency, 187 Osongsaengmyeong2-ro, Osong-eup, Heungdeok-gu, Cheongju-si 28159, Chungcheongbuk-do, Korea

**Keywords:** nonalcoholic fatty liver disease, Framingham steatosis index, physical activity, resistance training

## Abstract

Background: Nonalcoholic fatty liver disease (NAFLD) is the most common form of chronic liver disease. The present study aimed to investigate the association of NAFLD with leisure-time physical activity (PA) levels and resistance training (RT). Methods: We used data from large nationwide cohorts in Korea. NAFLD was defined based on the Framingham steatosis index. Participants were categorized into four groups based on RT frequency and adherence to PA guidelines (≥150 min/week of moderate-intensity PA): Low-PA, Low-PA+RT, High-PA, and High-PA+RT. Multiple logistic regression models were used to assess the risk of NAFLD according to leisure-time PA levels and regularity of RT. Results: When compared with Low-PA, High-PA decreased the risk of NAFLD by 17%, and High-PA+RT further decreased the risk by 30%. However, the additional reduction in risk associated with the addition of RT was observed in men (19%), but not in women. In the High-PA group, men had a significantly higher training frequency and period for RT than women. Conclusions: Following the PA guideline may confer protective effects against NAFLD, while adding RT to High-PA can further decrease the risk of NAFLD. Sex-based differences in NAFLD risk in the High-PA+RT group may be due to the differences in the frequency and period of RT.

## 1. Introduction

Nonalcoholic fatty liver disease (NAFLD) is characterized by the presence of abnormal lipid accumulation in the liver cells in the absence of significant alcohol consumption. The term is used to describe a wide range of fatty liver conditions, ranging from simple steatosis and nonalcoholic steatohepatitis (NASH) to significant liver diseases, such as cirrhosis and hepatocellular carcinoma [1]. NAFLD is also closely related to diabetes mellitus (DM), cardiovascular disease (CVD), and liver-related morbidity and mortality [2,3]. According a previous meta-analysis, the global prevalence of NAFLD is approximately 25%, with the highest prevalence in South America and the Middle East [4]. In addition, due to rapid economic growth and lifestyle changes across society, the prevalence of NAFLD in Asian countries has steadily increased. In particular, NAFLD prevalence in Korea has continuously increased over the past decade given increases in obesity/abdominal obesity, the adoption of sedentary lifestyles, and high fat intake [5], with meta-analytic estimates indicating a rate of 31% [6]. Therefore, there has been a growing interest in preventive and therapeutic strategies for NAFLD.

As there are currently no approved pharmacological strategies for the prevention or treatment of NAFLD, lifestyle modification including the addition of physical activity (PA) is the current recommended treatment. Several randomized controlled trials (RCTs) have demonstrated that PA exerts a protective effect on NAFLD [7,8,9], and epidemiologic studies have also reported an inverse correlation between increased PA and the risk of NAFLD [10,11,12]. Accordingly, the World Health Organization (WHO) has also recommended at least 150 min per week of moderate-intensity leisure-time PA to improve cardiorespiratory and muscular fitness and reduce the risk of chronic diseases such as DM, CVD, metabolic syndrome, and even cancer [13].

Resistance training (RT) is a type of leisure-time PA characterized by muscle contraction against external resistance, with the expectation of improving muscular fitness and physical function. The American College of Sports Medicine (ACSM) recommends performing RT 2 to 3 days per week in addition to aerobic exercise training (AT) to improve body composition, blood pressure (BP), and insulin resistance [14]. Research has indicated that RT significantly improves metabolic parameters, such as liver enzymes and intrahepatic lipids, at a lower exercise intensity and level of energy consumption than AT [15]. Thus, RT is likely to be more efficient than AT for patients with NAFLD, especially those with obesity or poor aerobic fitness. Although a previous study reported a significant association between RT and NAFLD in an Israeli cohort [16], the sample size was small, and further studies are required to verify the protective role of RT in the context of NAFLD. Furthermore, evidence regarding the combined effects of leisure-time PA levels and RT on NAFLD remains lacking, and potential differences among races and ethnicities remain to be explored using nationwide cohort data.

Therefore, in the present study, we aimed to investigate the association of NAFLD with leisure-time PA levels and regularity of RT using data from large nationwide cohorts. We further aimed to determine the effect of RT on NAFLD in participants with high leisure-time PA levels.

## 2. Materials and Methods

### 2.1. Study Participants

This cross-sectional study used data from the Korean Genome and Epidemiology Study (KoGES), which was conducted by the Korea National Institute of Health. The KoGES is a large consortium project consisting of six prospective cohort studies that aims to establish comprehensive healthcare guidelines for common complex diseases such as DM, hypertension, metabolic syndrome, obesity, CVD, and cancer [17]. For this study, we used 2003–2013 data from the KoGES_Health Examinee (HEXA) study, which included 173,202 urban residents aged 40–79 years, as well as data from the third wave of the KoGES_Ansan and Ansung study (2005–2006), which included 7515 participants aged 43–74 years living in these areas. Participants in each study underwent face-to-face surveys and physical examinations by trained medical staff. A detailed description of these cohort studies is provided in a previous study [17].

Among the 180,717 participants from the two cohorts, the following participants were excluded from the present study: those with a clinical history of CVD, hepatitis, liver cirrhosis, or any cancer (*n* = 15,643); those with incomplete data necessary for calculating the Framingham steatosis index (FSI) such as BP, triglycerides (TG), fasting blood glucose (FBG), aspartate aminotransferase (AST), or alanine aminotransferase (ALT) (*n* = 7483); those with no data regarding leisure-time PA levels (*n* = 5455); men and women with alcohol consumption >30 g/day and >20 g/day, respectively (*n* = 13,249; 10,946 men and 2303 women); and those who had no data available regarding covariates (*n* = 1332). Overall, 137,555 participants (96,948 women) were included in the final analyses (Figure 1). This study was approved by the Institutional Review Board Committee of the Korea National Institute of Health, Korea Disease Control and Prevention Agency (Approval No. 2021-04-02-P-A).

### 2.2. Definition of NAFLD

NAFLD was defined based on the FSI in patients without other liver-related diseases or significant alcohol consumption. Body mass index (BMI), TG, AST, ALT, hypertension status, and DM history were used to calculate the FSI according to the following formula [18]:

FSI = 1/(1 + exp(−x)) × 100, where x = −7.981 + 0.011 × age (years) − 0.146 × sex (female = 1, male = 0) + 0.173 × BMI (kg/m^2^) + 0.007 × TG (mg/dL) + 0.593 × hypertension (yes = 1, no = 0) + 0.789 × DM (yes = 1, no = 0) + 1.1 × ALT:AST ratio ≥ 1.33 (yes = 1, no = 0).

NAFLD was defined as an FSI ≥23 [18], and this diagnostic model has been externally validated in several independent cohorts including Korean patients [18,19,20].

### 2.3. Measurement of PA

All participants answered questionnaires providing details regarding their regular leisure-time PA and RT. For leisure-time PA, the intensity, frequency (per week), and duration (min) in a typical week were assessed. Participants were classified into two groups according to the WHO recommendation of at least 150 min per week of moderate-intensity leisure-time PA: Low-PA (not meeting the WHO recommendation) and High-PA (meeting the WHO recommendation) [13]. Moderate intensity activity was defined as participation in sports or exercise to the point of sweating. Regular RT was defined as participation in an RT program more than 1 day per week. Participants were divided into four groups based on leisure-time PA levels and regularity of RT: Low-PA, Low-PA+RT, High-PA, and High-PA+RT.

### 2.4. Covariates

Sociodemographic and health-related factors including age, sex, drinking and smoking habits, education level, hypertension, DM, BMI, waist circumference (WC), BP, and laboratory parameters were also considered in our analyses. Drinking and smoking habits were classified as “never,” “former,” and “current.” Education level was divided into elementary school graduate or lower, middle, or high school graduate, and college graduate or higher. Hypertension was defined as a previous diagnosis of hypertension, current use of antihypertensive drugs, systolic BP (SBP) ≥ 140 mmHg, or diastolic BP (DBP) ≥ 90 mmHg. DM was defined as a previous diagnosis of DM, current use of antidiabetic medications including insulin and oral hypoglycemic agents, FBG ≥ 126 mg/dL, or glycated hemoglobin (HbA1c) ≥ 6.5.

Anthropometric data including body weight, height, and WC were measured by trained healthcare providers using standardized protocols. BMI was calculated as body weight (kg) divided by height (m) squared (kg/m^2^). BP was also measured by trained healthcare providers using standard methods. SBP and DBP were defined as the average of two readings for the arm with the highest SBP obtained after 5 min of rest in a seated position. Blood samples were obtained after an overnight fast of at least 8 h. Biochemical assays were also performed for total cholesterol (T-Chol), high-density lipoprotein cholesterol (HDL-C), TG, FBG, AST, ALT, and HbA1c. A detailed description of the biochemical analysis is available elsewhere [17].

### 2.5. Statistical Analysis

All statistical analyses were conducted using SAS 9.4 software (SAS Institute, Cary, North Carolina, United States of America). Participant characteristics are presented as descriptive statistics. Continuous variables are presented as means ± standard deviations, while categorical variables are expressed as numbers and percentages (%). The chi-square test was used to compare sex ratios, drinking and smoking habits, education levels, participation in RT, and the prevalence of chronic diseases (e.g., hypertension, DM, and NAFLD) between groups. Independent t-tests were used to compare age, total time engaged in leisure-time PA (PA-time), BMI, WC, BP, T-Chol, HDL-C, TG, FBG, AST, ALT, and FSI between the groups.

A multiple logistic regression model was used to determine odds ratios (ORs) and 95% confidence intervals (CIs) for NAFLD prevalence. Models were adjusted for age, sex, drinking, smoking, education level, BMI, WC, T-Chol, ALT, hypertension, DM, and PA-time. Participants were categorized into one of four groups to investigate differences in the risk of NAFLD according to leisure-time PA levels, regularity of RT, and sex: Low-PA, Low-PA+RT, High-PA, and High-PA+RT. We also performed a subgroup analysis to examine differences in the risk of NAFLD according to RT and sex in those with High-PA levels (High-PA vs. High-PA+RT). All tests were two-tailed, and statistical significance was set at *p*-value < 0.05.

## 3. Results

Table 1 presents the general characteristics of study participants based on leisure-time PA levels and sex. Prevalence rates for NAFLD in our study population were 30.54% and 13.33% in men and women, respectively. The prevalence of NAFLD was low in physically active men and women (31.33% vs. 29.37% in men, 14.12% vs. 11.91% in women for Low-PA vs. High-PA, respectively). In both sexes, FSI values were also significantly lower in the High-PA group than in the Low-PA group (all *p* < 0.0001). The mean age was slightly higher in the High-PA group than in the Low-PA group. The prevalence of current drinking was also higher in the High-PA group, while the frequency of current smoking was lower. A low educational level (≤elementary school) was significantly more common in the Low-PA group than in the High-PA group for both men and women. Interestingly, the prevalence of hypertension and DM were relatively high in the High-PA group, as were rates of high SBP, DBP, and FBG. However, the Low-PA group exhibited relatively low HDL-C and high TG. Table 2 shows that high levels of leisure-time PA significantly decreased the risk of NAFLD after adjustment for covariates. Appendix A shows the inverse dose–response relationship between leisure-time PA levels and the risk of NAFLD after adjustment for covariates (all *p* for trend < 0.0001). Even though the prevalence of NAFLD was more than 50% lower in women than men, the protective effect of high leisure-time PA levels was similar in both sexes.

We further analyzed the additional effect of RT on NAFLD in participants with high or low leisure-time PA levels. The proportion of participants engaged in RT was 15.21% in men and 8.69% in women. The study participants were divided into four subgroups based on regularity of RT and leisure-time PA levels. The general characteristics of the four subgroups are presented in Appendix A. PA-time was significantly lower in the Low- and High-PA groups than in the Low-PA+RT and High-PA+RT groups, respectively. SBP, FBG, and T-Chol were lower, while HDL-C was higher, in the RT groups than in the non-RT groups. However, in the High-PA group, the prevalence of NAFLD significantly differed between the RT (17.91%) and non-RT (16.03%) subgroups.

The regression analysis also indicated that RT had no significant effect on NAFLD in the Low-PA group. However, High-PA was associated with a decreased risk of NAFLD regardless of RT (High-PA and High-PA+RT groups) (Table 3). Among both men and women, the High-PA+RT group had a significantly higher training frequency, a longer RT period, and a higher rate of long-term RT (≥1 year) than the Low-PA+RT group (all *p* < 0.0001). When we evaluated the additional effect of RT in the High-PA group (Appendix A), the beneficial effect was observed in men only (OR, 0.81; 95% CI, 0.73–0.90). Sex-based comparisons indicated that, in the High-PA+RT group, men had a significantly higher training frequency (*p* < 0.0001) and significantly longer training period (*p* < 0.0001) involving RT than women. In addition, the rate of participation in a long-term RT program (≥1 year) was significantly higher in men than in women in the High-PA+RT group (*p* < 0.0001).

## 4. Discussion

To our knowledge, the present study is the first to investigate the prevalence of NAFLD according to leisure-time PA levels and participation in RT in Korean adults using data from large nationwide cohorts. Our findings indicated that meeting the leisure-time PA recommendation (≥150 min/week of moderate-intensity PA) may be associated with protective benefits against NAFLD. More importantly, high levels of PA involving regular RT appears to be relatively more effective in preventing NAFLD. However, the additional effect of RT was observed only in men, which may have been due to differences in the regularity of RT (i.e., frequency and training period) between the sexes.

According to the recommendations of the WHO and ACSM, individuals should participate in moderate-intensity PA or sports for at least 150 min weekly to reduce the risk of chronic diseases such as hypertension, DM, CVD, and cancer [13,14]. NAFLD has become the most common chronic liver disease, and this is also closely related not only to liver-related morbidity but also to DM, metabolic syndrome, and CVD [2,3,21]. As for other chronic diseases, PA remains the recommended lifestyle modification for the management of NAFLD. Several recent studies have reported an inverse association between High-PA levels and the risk of NAFLD in Western countries such as the United Kingdom [12], the Netherlands [10], and the United States [11]. However, to the best of our knowledge, few epidemiological studies have focused on this association in Asian participants, especially in Korea, despite possible differences due to race- and ethnicity-based diversity. In our study, high levels of PA were associated with an approximately 20% reduction in the risk of NAFLD when compared with PA levels below the recommendation, after adjusting for other confounders. Our results are consistent with those from previous studies in Western countries [10,11,12].

While the prevalence of NAFLD was more than 50% lower in women than men, the protective effect of high leisure-time PA levels was relatively similar in both sexes in our study population. Several clinical studies have also demonstrated the beneficial effects of regular exercises on clinical findings in patients with NAFLD. In several RCTs, regular moderate-intensity AT (130–150 min/week of jogging or cycling) was effective in reducing intrahepatic TG, ALT, AST, body fat mass, and visceral lipid levels in patients with NAFLD [7,8,9]. In particular, beneficial effects were observed in patients with NAFLD following regular AT independent of weight loss [8,9]. Interestingly, however, our analysis indicated that the prevalence of hypertension and DM were relatively high in participants with high leisure-time PA levels. This may be related to increases in PA levels after being diagnosed with these diseases. Given that the present study was cross-sectional in nature, further longitudinal studies are required to determine the cause-and-effect relationships between PA levels and these diseases. Taken together, our results and those from previous RCTs indicate that increasing leisure-time PA to the level recommended by the WHO and ACSM is likely to be effective for managing and preventing NAFLD.

RT is also recommended for improving body composition, physical function, BP, and insulin resistance [14] and for decreasing the risks of mobility-related disability, hypertension, DM, CVD, and cancer [22,23,24]. Interestingly, regular participation in RT is more efficient in reducing liver enzyme levels and hepatic fat content in patients with NAFLD [8,25] than AT, at lower exercise intensities and levels of energy consumption [15]. Several studies have reported that, when compared with AT- or RT-only interventions, AT combined with RT exerts a synergistic effect to significantly improve biomarkers including insulin resistance, ALT, and T-Chol and lower all-cause mortality [26,27]. Furthermore, several meta-analyses and RCTs have demonstrated that regular RT can improve levels of intrahepatic lipids, ALT, AST, gamma-glutamyl transferase (GGT), T-Chol, TG, low-density lipoprotein cholesterol (LDL-C), FBG, insulin resistance, total fat, and trunk fat in patients with NAFLD [28,29,30,31]. However, to the best of our knowledge, few epidemiological studies have investigated the prevalence of NAFLD by considering both leisure-time PA levels and the regularity of RT. In our study, although both the High-PA and High-PA+RT groups exhibited a decreased risk of NAFLD, RT had no effect on NAFLD in the Low-PA groups. For both sexes, the training frequency and period of RT was higher in the High-PA group than in the Low-PA group. Given that a recent meta-analysis reported a dose–response relationship between training volume and muscular fitness gains [32,33], these differences in the frequency and training period of RT likely explain the significant difference in the risk of NAFLD between the groups (Low-PA+RT vs. High-PA+RT).

The subgroup analysis revealed a significant difference in the risk of developing NAFLD according to RT and sex in those with high PA levels. The risk of NAFLD was markedly lower in men of the High-PA+RT group than in those of the High-PA group; however, the difference was not significant in the female subgroups. To the best of our knowledge, only a few previous studies have investigated sex-based differences in the risk of developing NAFLD between High-PA and High-PA+RT groups. Among those with high levels of PA in our study, the frequency and training period of RT, as well as the rate of participation in a long-term RT program (≥1 year), were significantly higher among men than women. Although information regarding the intensity of RT was not available, these disparities in the regularity of RT are likely due to sex-based differences in NAFLD prevalence. A recent meta-analysis indicated that, when the same RT protocol was applied, men and women adapted similarly to RT in terms of both muscular strength and muscle hypertrophy [34]. Other meta-analyses have demonstrated that there is a graded dose–response relationship between RT frequency or weekly sets performed and muscular fitness gains, and these relationships seem to be related to increases in the total training volume [32,35]. Taken together, these results indicate that sex-based differences in the frequency and training period of RT are likely to create a significant gap in the risk of NAFLD between the sexes in those with High-PA levels. Thus, it may be necessary to encourage women to participate in regular RT or increase their training volume (e.g., frequency and training period) to reduce the risk of NAFLD. However, further RCTs are required to verify the sex-based differences in the synergistic effects of combined RT and high PA levels on the treatment for NAFLD by matching the total training volume and exercise intensity of the participants.

The major strength of our study was the use of large nationwide cohorts that are representative of the Korean general population aged 40–79 years, making our results generalizable to this population. Furthermore, to our knowledge, this is the first study to use data from large nationwide cohorts to investigate the risk of NAFLD while considering the combined effects of leisure-time PA levels and RT. However, there were some limitations to this study. First, although FSI has been externally validated in a community-based population in Korea using magnetic resonance imaging (MRI) [20], the actual prevalence of NAFLD may have been under- or overestimated. Second, data regarding leisure-time PA levels and regularity of RT were obtained using self-reported questionnaires, which may have introduced recall bias. Third, as this study involved a cross-sectional analysis, we were unable to deduce causal relationships given the nature of the study. Lastly, we were unable to obtain specific information regarding the intensity of RT given the self-reported nature of our questionnaires. When taken with previous results, our findings indicate that meeting the PA guideline and engaging in regular RT may exert a synergistic effect to aid in the management and prevention of NAFLD. However, further studies are required to determine the optimal intensity, frequency, and training period required to achieve this effect.

## 5. Conclusions

The present results suggest that following the leisure-time PA guideline (≥150 min/week of moderate-intensity PA) may be associated with protection against NAFLD, and that high leisure-time PA levels combined with RT are relatively effective in reducing the risk of NAFLD. Moreover, our analysis revealed a sex-based difference in the risk of NAFLD between the High-PA and High-PA+RT groups, which is likely due to differences in the regularity of RT (i.e., frequency and training period). Thus, further studies are required to determine whether there are sex-based differences in the synergistic effects of RT and high PA levels on the treatment and prevention of NAFLD. The results of our study are particularly relevant for reducing the risk of NAFLD in Korean adults.

## Figures and Tables

**Figure 1 ijerph-19-02350-f001:**
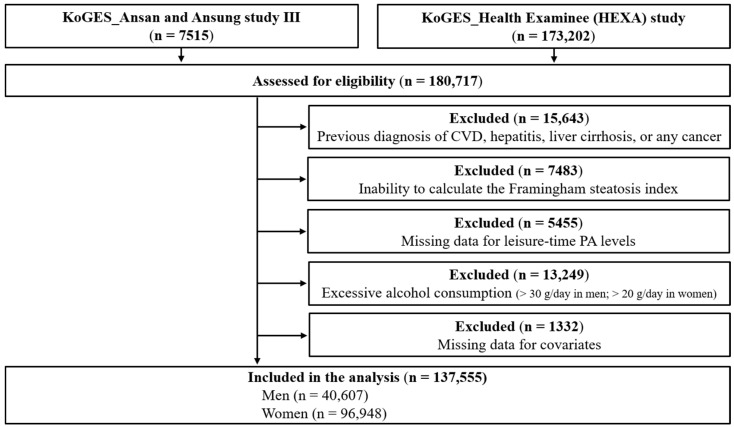
Flow diagram of participant inclusion and exclusion. CVD, cardiovascular disease; PA, physical activity.

**Table 1 ijerph-19-02350-t001:** Characteristics of study participants based on leisure-time PA levels.

Variables	Men (*n* = 40,607)		*p*-Value	Women (*n* = 96,948)		*p*-Value
	Low-PA (*n* = 24,239)	High-PA (*n* = 16,368)		Low-PA (*n* = 62,436)	High-PA (*n* = 34,512)	
**Age** (years)	53.25 ± 8.87	55.11 ± 8.61	<0.0001	52.43 ± 8.33	52.95 ± 7.67	<0.0001
**Education level**, *n* (%)			<0.0001			<0.0001
≤Elementary school	3407 (14.06)	1275 (7.79)		15,843 (25.38)	6392 (18.52)	
Middle/high school	12,801 (52.81)	8337 (50.93)		35,223 (56.41)	21,885 (63.41)	
≥College	8031 (33.13)	6756 (41.28)		11,370 (18.21)	6235 (18.07)	
**Drinking habit**, *n* (%)			<0.0001			<0.0001
Never drinker	6336 (26.14)	3793 (23.17)		42,828 (68.59)	22,851 (66.21)	
Ex-drinker	1881 (7.76)	1269 (7.75)		1252 (2.01)	652 (1.89)	
Current drinker	16,022 (66.10)	11,306 (69.08)		18,356 (29.40)	11,009 (31.90)	
**Smoking habit**, *n* (%)			<0.0001			<0.0001
Never smoker	7267 (29.98)	5229 (31.95)		60,210 (96.44)	33,665 (97.54)	
Ex-smoker	8486 (35.01)	7407 (45.25)		739 (1.18)	382 (1.11)	
Current smoker	8486 (35.01)	3732 (22.80)		1487 (2.38)	465 (1.35)	
**PA-time** (min/week)	20.64 ± 40.59	397.69 ± 265.87	<0.0001	17.70 ± 38.79	364.37 ± 218.41	<0.0001
**RT** (%)	2000 (8.25)	4176 (25.51)	<0.0001	2087 (3.34)	6342 (18.38)	<0.0001
**BMI** (kg/m^2^)	24.22 ± 2.81	24.44 ± 2.59	<0.0001	23.74 ± 3.07	23.66 ± 2.78	<0.0001
**WC** (cm)	85.50 ± 7.68	85.49 ± 7.23	0.93	78.98 ± 8.47	78.25 ± 7.86	<0.0001
**SBP** (mmHg)	124.35 ± 14.94	125.34 ± 14.64	<0.0001	120.52 ± 15.74	121.01 ± 15.58	<0.0001
**DBP** (mmHg)	78.26 ± 10.02	78.47 ± 9.62	<0.05	74.75 ± 9.92	74.93 ± 9.77	<0.01
**T-Chol** (mg/dL)	194.31 ± 34.26	193.75 ± 33.56	0.11	199.55 ± 35.76	200.67 ± 35.43	<0.0001
**HDL-C** (mg/dL)	48.00 ± 11.35	49.53 ± 11.65	<0.0001	55.17 ± 12.62	57.22 ± 13.10	<0.0001
**TG** (mg/dL)	152.01 ± 106.54	140.56 ± 96.60	<0.0001	116.43 ± 76.71	110.07 ± 70.46	<0.0001
**FBG** (mg/dL)	97.54 ± 24.25	98.64 ± 22.77	<0.0001	92.69 ± 19.26	92.98 ± 18.61	<0.05
**AST** (IU/L)	25.11 ± 13.25	24.90 ± 13.24	0.12	22.53 ± 22.20	22.54 ± 9.43	0.93
**ALT** (IU/L)	27.24 ± 21.18	25.49 ± 17.53	<0.0001	19.97 ± 23.21	19.45 ± 14.69	<0.0001
**Hypertension**, *n* (%)	7667 (31.63)	5836 (35.65)	<0.0001	15,769 (25.26)	9114 (26.41)	<0.0001
**DM**, *n* (%)	2621 (10.81)	2250 (13.75)	<0.0001	4300 (6.89)	2746 (7.96)	<0.0001
**FSI**	21.21 ± 20.10	20.22 ± 18.57	<0.0001	12.46 ± 14.10	11.60 ± 12.87	<0.0001
**NAFLD**, *n* (%)	7593 (31.33)	4807 (29.37)	<0.0001	8815 (14.12)	4109 (11.91)	<0.0001

PA, physical activity; PA-time, total time participating regularly in any sports or exercise to the point of sweating; RT, resistance training; BMI, body mass index; WC, waist circumference; SBP, systolic blood pressure; DBP, diastolic blood pressure; T-Chol, total cholesterol; HDL-C, high-density lipoprotein cholesterol; TG, triglycerides; FBG, fasting blood glucose; AST, aspartate aminotransferase; ALT, alanine aminotransferase; DM, diabetes mellitus; FSI, Framingham steatosis index; NAFLD, nonalcoholic fatty liver disease.

**Table 2 ijerph-19-02350-t002:** ORs for NAFLD according to leisure-time PA levels and sex.

	N	NAFLD (%)	Model 1 OR (95% CI)	Model 2 OR (95% CI)	Model 3 OR (95% CI)
**Total**	137,555	18.41			
Low-PA	86,675	18.93	1 (reference)	1 (reference)	1 (reference)
High-PA	50,880	17.52	0.82 (0.80–0.85) **	0.88 (0.85–0.91) **	0.80 (0.77–0.84) **
**Men**	40,607	30.54			
Low-PA	24,239	31.33	1 (reference)	1 (reference)	1 (reference)
High-PA	16,368	29.37	0.91 (0.87–0.95) **	0.85 (0.80–0.89) **	0.80 (0.75–0.85) **
**Women**	96,948	13.33			
Low-PA	62,436	14.12	1 (reference)	1 (reference)	1 (reference)
High-PA	34,512	11.91	0.79 (0.76–0.82) **	0.92 (0.87–0.96) *	0.83 (0.78–0.88) **

NAFLD, nonalcoholic fatty liver disease; PA, physical activity; OR, odds ratio; CI, confidence interval; BMI, body mass index; WC, waist circumference; T-Chol, total cholesterol; ALT, alanine aminotransferase; *, *p* < 0.001; **, *p* < 0.0001. Model 1 was adjusted for age and sex. Model 2 was adjusted for the variables in Model 1, plus drinking, smoking, education level, BMI, and WC. Model 3 was adjusted for the variables in Model 2, plus T-Chol, ALT, hypertension, and diabetes mellitus.

**Table 3 ijerph-19-02350-t003:** ORs for NAFLD according to leisure-time PA levels, regularity of RT, and sex.

	N	NAFLD (%)	RT Levels			OR (95% CI)
			Frequency	Training Period		
			(Days/Week)	(Month)	≥1 Year (%)	
**Total**	137,555	18.41				
Low-PA	82,588	18.91	-	-	-	1 (reference)
Low-PA+RT	4087	19.26	3.39 ± 1.71 ^a^	13.04 ± 24.12 ^a^	65.99 ^a^	0.94 (0.84–1.06)
High-PA	40,362	17.91	-	-	-	0.83 (0.79–0.86) **
High-PA+RT	10,518	16.03	4.36 ± 1.57 ^a^	18.59 ± 34.22 ^a^	81.98 ^a^	0.70 (0.65–0.76) **
**Men**	40,607	30.54				
Low-PA	22,239	31.44	-	-	-	1 (reference)
Low-PA+RT	2000	30.00	3.49 ± 1.81 ^a^	15.33 ± 31.06 ^a^	72.80 ^a^	0.94 (0.82–1.08)
High-PA	12,192	30.24	-	-	-	0.84 (0.79–0.90) **
High-PA+RT	4176	26.82	4.44 ± 1.70 ^a^	21.95 ± 44.04 ^a^	85.56 ^a^	0.67 (0.61–0.75) **
**Women**	96,948	13.33				
Low-PA	60,349	14.30	-	-	-	1 (reference)
Low-PA+RT	2087	8.96	3.29 ± 1.59 ^a^	10.85 ± 14.34 ^a^	59.46 ^a^	0.89 (0.73–1.10)
High-PA	28,170	12.58	-	-	-	0.83 (0.78–0.88) **
High-PA+RT	6342	8.92	4.30 ± 1.48 ^a^	16.38 ± 25.56 ^a^	79.63 ^a^	0.80 (0.71–0.90) *

NAFLD, nonalcoholic fatty liver disease; PA, physical activity; RT, resistance training; OR, odds ratio; CI, confidence interval; BMI, body mass index; WC, waist circumference; T-Chol, total cholesterol; ALT, alanine aminotransferase; ^a^, *p* < 0.0001 compared Low-PA+RT with High-PA+RT; *, *p* < 0.001; **, *p* < 0.0001. Adjusted for age, sex, drinking, smoking, education level, BMI, WC, T-Chol, ALT, hypertension, and diabetes mellitus.

## Data Availability

The data in this study were from the Korean Genome and Epidemiology Study (KoGES; 4851-302), Korea National Institute of Health, Korea Disease Control and Prevention Agency, Korea.

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
