# Peer review of "Protective Effect of Leisure-Time Physical Activity and Resistance Training on Nonalcoholic Fatty Liver Disease: A Nationwide Cross-Sectional Study"

_ijerph, 2022, doi:10.3390/ijerph19042350_

Round 1
Reviewer 1 Report
In their comprehensive study Park et al. analyzed the protective effect of leisure-time physical activity and resistance training regularity on NAFLD risk in large nationwide cross-sectional study from Korea.
Here,137,555 study participants were categorised into four different groups (low-physical activity (defined as <150min/week),” “low-phyiscial activity + resistance training, “high-PA (defined as >150min/week)”,and “high-physical activity + resistance training“ and then the risk of NAFLD was calculated by using different multiple logistic regression models.
Medical as well as questionnaire data from the Korean Genome and Epidemiology Study (KoGES), which was conducted by the Korea National Institute of Health were used for this purpose.
The authors were able to observe that high levels of physical activity reduced the risk of NAFLD by 17%, and high-physical activity together with regular resistance training further decreased the NAFLD risk by 30%, when compared to study participants who only performed low amounts of physical activity (<150min/week). Interestingly, the observed risk reduction with additional resistance training was observed only in males.
While the study is of interest and is based on data from a large cohort (> 100,000 study participants), from the reviewer's point of view, several aspects need to be revised and better addressed before publication can be considered.
Major comments:
From the reviewer's point of view, it should be more highlighted what the “novelity“ of these study data is and it needs to be better decribed what differentiates this study from the previous epidemiological studies focusing on the association of PA/RT and NAFLD in Asian participants.
There are already comprehensive data from randomized controlled trials which have demonstrated that both aerobic and resistance training significantly reduces the risk of NAFLD and the NAFLD-associated cardiovascular risk.
Data on the severity of NAFLD should be added.
How many patients had simple steatosis? How many patients had NASH and or progressive fibrosis (≥F2)?
In this regard, information on the Fibrosis-4 (FIB-4) and the NAFLD Fibrosis Score (NFS) should be implemented.
The authors focus primarily on the “ regularity“ of resistance training. Here they describe that “regularity of RT was determined according to whether the individuals participated regularly in the RT program more than once a week.“
Can conclusions also be drawn about effects of the intensity of resistance training?
When looking at the amount of physical activity, the differences are dramatic.
Physicial activity values of 20.64 vs. 397.69 min/week and 17.7 vs. 364.37 min/week are reported for men and women respectively. Thus, the active participants move about 20x-times more than the inactive ones.
From the reviewers' point of view, here, two maximum extremes are compared: a maximal sedentary behaviour group and a group that is extremely active, which of course produces significant differences.
Wouldn't it be better to divide the amount of physicial exercise into several degrees, for example low (<75min/week), moderate (75-150 min/week), high (150-250 min/week) and very high (>250 min/week)? Also in view of assessing whether the recommended 150 min/week is sufficient as a recommendation or even more would be more beneficial?
Minor comments:
Introduction:
It should be briefly discussed why the prevalence of NALFD in Korea is higher than the global average?
It should be briefly described, which measures can be summarized under the term “resistance training“.
Methods:
The “Framingham steatosis index (FSI)“ were used to identify peaple with NAFLD.
Why was this score preferred to other established algorithms, such as the Fatty liver index (FLI), which is also validated in the Asian region?
Results:
Interestingly, the prevalence of hypertension and type 2 diabetes were significantyl higher in study participants with high-physical activity levels.
What could be the reasons for this? This observation should be discussed and described more extensively.
Discussion:
The authors described that in several previous studies AT and RT, relative to AT- or RT-only intervention, showed synergistic effects in improving metabolic biomarkers, insulin resistance, and also in lowering all-cause mortality.
Is information also available on the effect on the incidence of major adverse cardiovascular outcomes in the group of NAFLD patients? If so, this should be addressed as cardiovascular events represent a common cause of mortality in this risk population.
Author Response
We would like to thank the reviewers for their positive reviews and comments regarding our study. Please see the attachment. We have responded to each point. Changes to the revised manuscript are marked in red font.

Reviewer 2 Report
In their manuscript, Park et al. investigate the effects of exercise on NAFLD in a crosssectional study from 2004 to 2013 consisting of 173,202 urban residents aged 40–79 years old and another cohort study of 7,515 participants aged 43–74 from 2005 to 2006 depicting the general population in Korea. NAFLD was defined by the FSI without other liver-related diseases, and prevalence rates for NAFLD were 30.54% and 13.33% in males and females, respectively. A questionnaire captured leisure time activity and resistance training.
I fully acknowledge the author's efforts to provide clinically relevant information for NAFLD patients in Korea that could potentially lead to increased awareness of exercise effects on NAFLD development. The study is well written, timely, cites all necessary studies, and is well presented. Still, I have some comments that need to be addressed:
- The authors acknowledge the limitations of the FSI score. Still, I am wondering if ICD10 codes were also available in this cohort? It would be fascinating to see how many of those participants had a diagnosis of NAFLD and how the prevalence of NAFLD ICD10 code changed over the years as the awareness of NAFLD increased.
2. Moreover, I wonder if the authors can see a linear or a cubic relationship between leisure-time activity and NAFLD development. Splines could depict the relationship between excericse minutes and NAFLD development, which would significantly enhance the value of the study. In addition, it would be interesting to see the optimal cutoff for exercise in this study to prevent NAFLD and if it is close to the 150 min the author's use as a reference. It would also be of great value if the optimal cutoff differed between men and women, which I would expect after the results of your study.
- I encourage the authors to add ORs below table 1 (or in a different table) to show the effects of exercise on all categorical covariates. Moreover, I encourage the authors to add % of participants above the upper limit of the norm for all longitudinal covariates and the corresponding OR to table 1. For example, I would find it fascinating to see the OR for ALT elevation of triglyceride elevation, and this could give insights into other effects of exercise besides NAFLD protection. Moreover, it would be great to add and extend the cohort's percentages of cardiovascular/metabolic diseases.
- I encourage the authors to add Fib4 to table 1 and show the effects of exercise on Fib4.
Minor comments
- The title underplays the importance of the study. The authors might consider an alternative title that attracts readers' attention more. In particular, the phrase "Resistance Training Regularity" is unclear. In my opinion, it is more about the amount of resistance training than the regularity.
- I encourage the authors to carefully revise the grammatic structure of the manuscript as, throughout the manuscript, several grammatical phrases occur that are unclear. See examples only from the abstract below:
- "Multiple logistic regression 19 model was used to assess the risk of NAFLD."
- "However, the risk reduction 21 with additional RT was observed only in males (19%) "– Here, the comparison is missing.
Author Response

(The authors gave the same response as above.)

Round 2
Reviewer 1 Report
The authors carefully revised the manuscript and adequately addressed the reviewer's questions.
Author Response
We would like to thank the reviewer for your positive reviews and comments regarding our study, and good luck with your future. We will do our best until the final publication.
Reviewer 2 Report
Unfortunately, neither ICD10 codes nor Fib4 score are available, but the authors sufficiently addressed all critical points that I have raised before.
Moreover, two additional supplementary Tables have been added. The newly presented data strengthens the relevance of physical activity for NAFLD. For the Table that has been generated to show the effect of activity on diabetes, I encourage the authors to add it as a Supplemental table, as diabetes mellitus is a crucial risk factor for NAFLD and it might be interesting to discuss if the protective effect for NAFLD might be due to the reduction of diabetes risk in this population.
Author Response
We would like to thank the reviewer for your positive reviews and comments regarding our study. We have responded to each point. Please see the attachment.
